# Epigenetic Mechanisms Underlying Melanoma Resistance to Immune and Targeted Therapies

**DOI:** 10.3390/cancers14235858

**Published:** 2022-11-28

**Authors:** Andrey Rubanov, Pietro Berico, Eva Hernando

**Affiliations:** 1Department of Pathology, NYU Grossman School of Medicine, New York, NY 10016, USA; 2Interdisciplinary Melanoma Cooperative Group, Perlmutter Cancer Center, NYU Langone Health, New York, NY 10016, USA

**Keywords:** melanoma, epigenetics, therapy resistance, viral mimicry, phenotype switching

## Abstract

**Simple Summary:**

Despite major recent therapeutic advances, melanoma remains the deadliest form of skin cancer due to the capacity of melanoma cells to adapt to drug treatment and become resistant. Improved understanding of melanoma suggests that it resists treatment not just due to DNA changing mutations but also due to changes in DNA accessibility with respect to the reading and creation of different proteins. Here, we summarize the various ways in which different DNA regions become more or less open, impacting patient response to anti-cancer therapies. For instance, targeting specific proteins governing the expression of viral-like genes can alert the immune system and enhance the effects of anti-cancer therapies. Alternatively, changes in DNA accessibility allow the activation of alternative survival signals that allow melanoma cells to escape death upon treatment. Targeting factors modulating DNA accessibility in cancer cells has recently become possible through technological developments. These novel therapies, administered alone or in combination with existing ones, represent the next frontier in the treatment of advanced melanoma.

**Abstract:**

Melanoma is an aggressive skin cancer reliant on early detection for high likelihood of successful treatment. Solar UV exposure transforms melanocytes into highly mutated tumor cells that metastasize to the liver, lungs, and brain. Even upon resection of the primary tumor, almost thirty percent of patients succumb to melanoma within twenty years. Identification of key melanoma genetic drivers led to the development of pharmacological BRAF^V600E^ and MEK inhibitors, significantly improving metastatic patient outcomes over traditional cytotoxic chemotherapy or pioneering IFN-α and IL-2 immune therapies. Checkpoint blockade inhibitors releasing the immunosuppressive effects of CTLA-4 or PD-1 proved to be even more effective and are the standard first-line treatment. Despite these major improvements, durable responses to immunotherapy and targeted therapy have been hindered by intrinsic or acquired resistance. In addition to gained or selected genetic alterations, cellular plasticity conferred by epigenetic reprogramming is emerging as a driver of therapy resistance. Epigenetic regulation of chromatin accessibility drives gene expression and establishes distinct transcriptional cell states. Here we review how aberrant chromatin, transcriptional, and epigenetic regulation contribute to therapy resistance and discuss how targeting these programs sensitizes melanoma cells to immune and targeted therapies.

## 1. Introduction

Representing only 1% of skin cancer cases but accounting for over 80% of deaths, melanoma is the deadliest form of skin cancer [1]. Ultraviolet exposure [2] of neural-crest-derived epidermal melanocytes [3] (or melanocyte stem cells [4,5]) generates highly mutated primary tumors [6,7,8] capable of rapidly metastasizing to the liver, lungs, and brain, even after complete surgical resection [9]. Half of melanoma patients carry conformation-changing BRAF^V600E^ mutations [10,11] which constitutively activate the mitogen-activated protein kinase (MAPK) pathway. BRAF^V600E^ inhibitor vemurafenib (FDA-approved in 2011) increased objective response rates (ORRs) from 5% to 50% compared to dacarbazine chemotherapy, and improved 6-month overall survival (OS) from 64% to 84% [12]. Following FDA approval of the MEK inhibitors cobimetinib and trametinib, combinatorial BRAF inhibition (BRAFi) and MEK inhibition (MEKi) increased 5-year OS to 34% [13] from 7% with dacarbazine plus interferon alpha (IFN-α) [14]. Targeted MAPK therapy recast metastatic melanoma from a rapid death sentence to a treatable condition with favorable outcomes for many patients.

Unfortunately, of the 40–60% eligible (BRAF^V600E^) patients, half of targeted therapy recipients display intrinsic or develop acquired resistance [15,16]. The intratumoral heterogeneity characteristic of melanoma gives rise to coexisting subclones with varying sensitivities to targeted therapy [17,18]. Intrinsic and acquired resistance are predominantly explained by MAPK re-activation [17]. Upstream of BRAF, gain-of-function mutations in NRAS and receptor tyrosine kinases (AXL, EGFR, IFG-1R, KIT, MET, and PDGFR) [19,20,21] or loss-of-function mutations in the tumor suppressor NF1 [22,23] hinder the efficacy of BRAFi and MEKi. Downstream of BRAF, MEK1/2 gain-of-function mutations [24,25] and increased levels of COT1, a MAPK kinase activating MEK1, also enable therapy resistance [26]. Additionally, resistance frequently involves activation of the PI3K/AKT/mTOR survival pathway via loss of the tumor suppressors PTEN and RB1 [27]. Similarly, acquired resistance typically occurs via delayed re-activation of MAPK (or analogous survival pathways), RAF isoform switching [28], BRAF copy number amplification [29], and alternative splicing isoforms [30]. 

First-generation immune therapy based on broad immune activation via IFN-α or IL-2 proved ineffective or exceedingly toxic [31,32]. Immune checkpoint blockade emerged chronologically parallel with targeted therapy, pioneered by the Nobel Prize-granted discovery of cytotoxic CD8^+^ T cell inhibition by CTLA-4 [33]. T cell surface marker CTLA-4 outcompetes the lower-affinity co-stimulatory receptor CD28 for binding to CD80 and CD86 on antigen-presenting cells (APCs), deactivating T cells via IL-2 inhibition and cell-cycle arrest [34]. FDA approval of ipilimumab, a monoclonal human CTLA-4 antibody, unlocked a novel approach to cancer treatment and improved patient outcomes [35], further potentiated by addition of anti-PD-1 antibodies [36,37]. Binding of T cell surface marker PD-1 to ligand PD-L1 inhibits T cell function via de-phosphorylation of the T cell receptor (TCR) [34]. While CTLA-4 inhibits early priming events between APCs and TCRs in the lymph nodes, PD-1 restrains activated T cells in peripheral tissues [34]. However, despite the monumental success of checkpoint blockade, half of treated patients fail to achieve long-term benefits [38]. Targeting of the alternative immune checkpoints LAG-3, TIM-3, and TIGIT has been explored [39,40]. However, despite recent FDA approval of combinatorial LAG-3 and PD-1 treatment [41], clinical progress has stagnated. A plateau in responses along with a lack of robust biomarkers for patient selection warrants a search for molecular mechanisms facilitating intrinsic and acquired resistance.

Intrinsic resistance affects half of anti-PD-1 and over 70% of anti-CTLA-4 immunotherapy recipients. Tumor-inherent factors facilitating intrinsic resistance include alterations in interferon signaling within the tumor and its microenvironment, reduced antigenicity and immunogenicity, and disrupted T cell infiltration and activity [42]. Tumor neoantigens matching patient TCRs represent a key component of tumor immune clearance in checkpoint blockade responders [43,44]. High tumor mutational burden precedes increased neoantigen generation [45] and is significantly associated with but not predictive of clinical treatment benefit [46,47,48]. Unsurprisingly, as a regulator of antigen presentation, interferon gamma (IFN-γ) signaling has been implicated in immunotherapy resistance. Binding of IFN-γ to its tumoral receptors induces inhibitory PD-L1 expression via JAK-STAT signaling [49]. Genetic defects in IFN-γ curb PD-L1 expression and suppress the efficacy of PD-1 immunotherapy [50]. However, clinical trials combining JAK inhibitors with PD-1 blockade have been generally unsuccessful [42,51]. Furthermore, tumors insensitive to IFN-γ escape immune surveillance via downregulation of MHC class I antigen presentation [52,53]. Unfortunately, the epigenetic alterations producing active IFN signaling cell states are still poorly understood. 

Acquired resistance to immune checkpoint blockade prevents durable responses through mechanisms analogous to intrinsic resistance. Two fundamental models are proposed [54]. The first describes Darwinian selection of resistant clones present from treatment onset (as a consequence of intratumoral heterogeneity) following depletion of sensitive clones with treatment. The second, homeostatic resistance, suggests the emergence of resistant clones due to the selective pressure imposed by treatment. Although the primary mode remains unknown, the underlying mechanisms are strikingly similar to intrinsic resistance and include lack of neoantigens, impaired antigen presentation, diminished IFN signaling, cellular plasticity defined by de-differentiation, epithelial-to-mesenchymal (EMT) transition [55,56], and WNT activation [57,58]. Recent studies have focused on the contribution of epigenetic regulators to resistance due to their ability to alter global transcriptional networks and produce resistant cell states.

## 2. Role of the Epigenome in Therapeutic Resistance

Epigenetic alterations enable more rapid and reversible modulation of cellular responses than changes to the coding genome. Melanoma tumorigenesis and progression has been linked to dysregulation of epigenetic mechanisms, such as chromatin remodeling complexes (INO80 [59,60], ISWI [61,62], and SWI/SNF [63,64,65]); histone post-translational modifications (PTMs) by histone acetyltransferases (HATs) [66], deacetylases (HDACs) [67], methyltransferases (HMTs) [68], and demethylases (HDMs) [69]; histone variants [70,71,72]; DNA [73,74,75] and RNA [76,77,78] methylation; plus non-coding [79,80,81,82], micro- [83,84,85], and circular [86,87,88] RNA. Unsurprisingly, epigenetic changes are also linked to immune [89,90,91,92,93,94,95,96,97] and targeted [98,99,100,101,102,103] therapy resistance. Therapy resistance develops rapidly and is unlikely to be driven by de novo mutations [17,104]. Rather than undergoing an irreversible and time-consuming mutagenic process, melanoma cells respond to microenvironmental signals and/or stochastic intracellular fluctuations to remodel epigenetic landscapes and adopt reversible gene expression programs. 

For instance, decreased PD-L1 expression and melanoma treatment-resistance is associated with global hypermethylation [105] and is reversible with DNA methyltransferase (DNMT) inhibitors [106]. Interestingly, DNMT inhibitors sensitized a pre-clinical melanoma model to CTLA-4 immunotherapy [107] by de-repressing endogenous retroviruses (ERVs) and activating an IFN response. Additionally, cyclin-dependent kinase 9 (CDK9) inhibition with toyocamycin activated the SWI/SNF catalytic subunit BRG1 and synergized with DNMT inhibition to de-repress endogenous retroviral elements and interferon signaling, sensitizing an in vivo ovarian cancer model to PD-1 immunotherapy [108,109]. Here we examine recent studies advancing our understanding of epigenetic regulation of ERVs and the emerging role of viral mimicry in overcoming melanoma immunotherapy resistance. Furthermore, targeted therapy resistance is not entirely explained by the acquisition of further mutations or genetically resistant clones but rather phenotypic alterations of cell states [110,111,112,113]. Interconnected oncogenic networks enable melanoma cells to circumvent MAPK inhibition via upregulation of survival pathways, molecular alteration of targets, or downregulating target expression. Here, we explore the role of epigenetic modulators in activating survival pathways and the contribution of phenotype switching to targeted therapy resistance.

### 2.1. Interferon-Mediated Viral Mimicry Facilitated by ERV De-Repression Enhances Response to Immunotherapy

Endogenous retroviral elements are genomically integrated fossil records of former retroviral infections in humans. Characterized as autonomous retrotransposons, ERV sequences occupy up to 8% of the human genome [114] but are often epigenetically repressed [115,116]. Human ERVs (HERVs) have been classified into over twenty families primarily on the basis of their tRNA-binding specificity [117]. While most HERVs exist defectively as long terminal repeats (LTRs), some HERV-K (HML2) members possess open reading frames (ORFs) for *gag*, *pol*, *env*, *Rec* [118], and *Np9* [119] and produce viral particles [120,121]. A regulatory loop between *Rec* and MITF inhibits epigenetic phenotype switching between proliferative and invasive melanoma states [122]. Additional ERVs have been co-opted by mammalian genomes in regulating cellular immunity [123,124]. HERV-K mRNA is expressed in primary and metastatic melanoma [125,126] and generates viral proteins containing antigenic and therapeutically targetable epitopes [127,128]. ERV retention of viral features endows immunogenicity, triggering immune responses [124,127] and activating viral mimicry [129]—a phenomenon of beneficial cellular hypochondria characterized by a viral-like immune response in lieu of infection. Driven by interferon signaling following aberrant dsRNA accumulation, viral mimicry is gathering momentum as a major mechanism of immunotherapy response [130,131].

Recent pre-clinical studies have demonstrated significantly improved melanoma immunotherapy responses due to viral mimicry. The PRMT family of methyltransferases, commonly upregulated in cancer [132], facilitate methyl deposition on arginine residues. PRMTs regulate antiviral responses [133,134] through mitochondrial antiviral-signaling-protein (MAVS)-dependent interferon and proinflammatory responses [135]. PRMT7 inhibition induced viral mimicry and sensitized therapy-resistant B16F10 melanoma tumors to CTLA-4 and PD-1 treatment in vivo [136]. Pharmacological inhibition or CRISPR/Cas9-mediated knockout of PRMT7 hypomethylates ERVs through DNMT silencing and leads to dsRNA accumulation—a common characteristic of viral infections. In addition, PRMT7 repressively bi-methylates H4R3 residues in the RIG-I and MDA5 promoters. Their increased transcription following PRMT7 knockout enhances dsRNA detection, activating IFN and interferon-stimulated gene (ISG) transcription through MAVS. This enhanced immunogenicity, antigenicity, and CD8^+^ T cell infiltration, improving responses to immunotherapy. Unfortunately, efforts to inhibit the PRMT family, including Phase I clinical trials, have focused on Type I and II PRMTs, leaving PRMT7 (Type III) behind [137]. SGC3027, used in this study, was recently reported as the first PRMT7-specific pharmacological inhibitor [138], but clinical testing has not been conducted yet.

Genetic or pharmacological inhibition of histone demethylase LSD1 (also known as KDM1A) also stimulates ERV expression in melanoma [139]. Intriguingly, LSD1 ablation resulted in decreased protein levels of DICER, AGO2, and TRBO2—key components of the dsRNA-recycling RISC complex. In addition to physically interacting with RISC, AGO2 protein stability is diminished upon LSD1 knockout via demethylation of the AGO2 K726me1 residue. Detected by upregulated TLR3 and MDA5, supraphysiological dsRNA levels plus reduced RISC activity induced IFN-β signaling. Consequentially, upregulation of MHC-I complexes and PD-L1 expression sensitized treatment-resistant B16F10 melanoma cells to PD-1 checkpoint blockade due to increased CD4^+^ and CD8^+^ T cell infiltration. At least half a dozen LSD1 inhibitors have gone through or are currently undergoing clinical assessment for oncological and neurological treatments; however, none have reached Phase III yet [140].

A string of recent publications have validated histone methyltransferase SETDB1 as an important regulator of melanoma immune responses. Initially, SETDB1 loss was demonstrated to de-repress transposable elements (TEs) resembling ERVs, immune gene clusters, and MHC-I loci [141]. Surprisingly, viral mimicry was not observed due to lack of bi-directional transcription and interferon signaling, despite the emergence of accessible interferon gene clusters following SETDB1 knockout. However, due to the presence of long terminal repeats and intact open reading frames containing viral *gag*, *pol*, and *env* genes, de-repressed TEs encoded immunogenic peptides predicted to bind MHC-I complexes. Augmented antigenicity enhanced infiltration of T cells carrying corresponding TCRs, sensitizing treatment-resistant B16F10 cells to in vivo PD-1 and CTLA-4 checkpoint blockade (however, these effects have only been tested in GVAX melanoma cells expressing GM-CSF [142]). 

Shortly after, it was demonstrated that histone demethylase KDM5B recruits SETDB1 to silence ERVs in a demethylase-independent manner [143]. KDM5B-knockout melanoma cells treated with proteasome inhibitor MG132 significantly increased SETDB1 levels, suggesting a protective role for KDM5B in proteasome-dependent degradation of SETDB1. Surprisingly, KDM5B loss (with concomitant SETDB1 reduction) increased bi-directional transcription of ERVs (such as MMVL30), accumulating dsRNA stress, activating type I interferon (IFN-I) signaling, upregulating MHC-I signaling, and inducing ISG expression. Accordingly, KDM5B knockout sensitized treatment-resistant YUMM1.7 melanoma cells to in vivo anti-PD-1 treatment through boosted CD8^+^ T cell activity. Corroborating the direct involvement of the IFN-I response, depletion of cytosolic DNA sensor cGAS or RNA sensor MDA5 ablated ISG expression and rescued YUMM1.7 in vivo tumor growth.

Finally, knockout of chromatin regulator ATF7IP or its interacting partner SETDB1 augments immunogenicity due to ERV expression and mRNA intron retention [144]. ATF7IP knockout induces interferon signaling through IRF7 and IRF9 upregulation, enhancing tumor immunogenicity and anti-tumor immunity via increased T cell activity. In MC38 colon carcinoma and KP lung carcinoma cell lines, ATF7IP knockout inhibited spliceosome activity, increasing intron expression and producing antigenic neoepitopes. Although no SETDB1-specific inhibitors have yet been developed, several histone lysine methyltransferase inhibitors have been tested in multiple pathological settings [145]. Recently and relevantly, Mithramycin A and Mithralog EC-8042 were shown to inhibit melanoma SETDB1 levels and improve the efficacy of MAPKi treatment [146]. KDM5B, upregulated in aggressive phenotypes of multiple cancer types, has been the target of numerous pharmacological inhibitors, but none have advanced to the clinic due to off-target effects and toxicity [147,148].

Viral mimicry defines a cell state characterized by a viral immune response in absence of infection. Induction of an anti-viral gene signature, driven by interferon signaling, activates MHC-I and PD-L1 expression (Figure 1). The resultant antigenicity alerts host immunity and reduces tumoral immune evasion. Despite some discrepancies in mechanisms and effector genes, LSD1, PRMT7, SETDB1, KDM5B, and ATF7IP functionally converge to evade melanoma immune clearance via repression of ancient genomic ERV elements that induce interferon signaling and translate immunogenic peptides for MHC-I presentation to CD8^+^ T cells (Table 1). Thus, inhibition of these immune-suppressive mechanisms could synergize with clinical immune checkpoint blockade.

### 2.2. Targeted Therapy Resistance via Epigenetic Upregulation of Survival Pathways Parallel to MAPK Signaling

Epigenetic mechanisms induce melanoma resistance to BRAF and MEK inhibitors, often by upregulating survival pathways functionally analogous to MAPK signaling, as illustrated by the following examples (Table 2).

BMI1 is a transcriptional repressor within the polycomb repressive complex 1 (PRC1) that mono-methylates H3K27, enabling PRC1 binding, and inhibiting transcription factor access via chromatin compaction [149]. BMI1 overexpression induced resistance in melanoma cells sensitive to pharmacological BRAF^V600E^ inhibition, while BMI1 silencing increased BRAFi sensitivity proportional to the degree of knockdown. Upregulation of both WNT5a and its receptor ROR2 in BMI1-overexpressing persister cells revealed the contribution of WNT signaling to BRAFi sensitivity. In fact, treatment sensitivity was also proportional to the degree of WNT5a knockdown. In medulloblastomas, BMI1 modulation of receptor tyrosine kinases in the MAPK pathway altered patient responsiveness to MEK inhibition [150]. BMI1 upregulation in multiple cancer types and a described role in cancer stem cells (CSCs) has led to the development of pharmacological inhibitors. BMI1 hyperphosphorylation by the small molecule PTC596 impairs protein function [151]. Recent Phase I results indicate that orally bioavailable PTC596 has a tolerable human safety profile and is pre-clinically efficacious in mice as a monotherapy against leiomyosarcomas and glioblastoma, supporting its further development [152]. This is exciting, considering that clinical trials targeting the polycomb complex have to date focused solely on EZH2, the catalytic domain of polycomb repressive complex 2.

Activation of the PI3K/AKT/mTOR pathway facilitates melanoma resistance to BRAF and MEK inhibition. A CRISPR-Cas9 sgRNA knockout screen targeting chromatin factors in BRAF^V600E^ melanoma cells in the presence of MAPKi rendered histone acetyltransferase (HAT) and deacetylase (HDAC) enzymes as hits [153,154]. Intriguingly, histone deacetylase SIRT6 haploinsufficiency promoted melanoma MAPKi resistance, while complete loss conferred sensitivity due to induction of the DNA damage response. SIRT6 haploinsufficiency resulted in increased H3K56 acetylation at the IGFBP2 locus, increasing chromatin accessibility and IGFBP2 expression. IGFBP2 activates IGF-1 receptor (IGF-1R), which triggers PI3K/AKT/mTOR survival signaling, enabling cell persistence in the presence of BRAF and MEK inhibition (MAPKi). Treatment with IGF-1R inhibitor linsitinib, which prevents IGF-1R autophosphorylation, overcomes SIRT6 haploinsufficiency resistance in vitro and in vivo. Supporting these findings, the authors showed that IGFBP2 transcript and protein levels were associated with poor prognoses for primary melanoma patients. Single-agent cixutumumab, a monoclonal antibody targeting IGF-1R, was tested on eighteen patients with metastatic uveal melanoma and exhibited low toxicity but only incomplete or partial responses, requiring further combinatorial studies with MAPK inhibitors to assess its utility [155]. IGF-1R inhibition has received abundant attention over the last decade for multiple cancer types, with the first FDA IGF-1R inhibitor, teprotumumab, a monoclonal antibody indicated for autoimmune Graves’ orbitopathy [156], approved in 2022. Further pre-clinical characterization may uncover a benefit in resistance delay or prevention by combining IGF-1R and MAPK inhibition in BRAF^V600E^ patients with high IGFBP2 expression. Complete SIRT6 loss results in hyperacetylation and global chromatin disarray, also enhancing MAPKi sensitivity. SIRT6 inhibitors have not reached the clinic, but numerous small-molecule modulators are available [157]. Four HDAC inhibitors are currently FDA-approved as cancer therapeutics and represent the most clinically successful epigenetic regulatory mechanisms, with almost two dozen completed, ongoing, or recruiting trials in melanoma alone [158]. Unfortunately, benefits are observed only in a minority of patients, in combination with other treatments, and are typically quickly followed by resistance and disease progression.

The WNT, PI3K/AKT/mTOR, and IGF signaling pathways have been shown to play key roles in melanocytes and the surrounding stroma throughout various developmental stages [159,160,161]. Accordingly, their dysregulation is associated with multiple stages of malignancy, from initiation to metastasis. Accumulating both a favorable and sufficient quantity of oncogenic mutations in these survival pathways by random chance is a time-intensive process. Epigenetic modulation facilitates complex changes in cell states without alterations in coding sequences (Figure 2). Shifting chromatin accessibility in multiple genomic loci enables the expression of multiple signaling components simultaneously—a necessary feature for sufficient pathway activation.

### 2.3. Phenotype Switching and Targeted Therapy Resistance

Cellular plasticity enables tumor cells to respond to stress and has been implicated as a mechanism of therapy resistance. Melanoma cells, regardless of their genetic subtype, are not truly epithelial or mesenchymal and rapidly switch between different phenotypes through EMT/MET-like mechanisms [162]. Accordingly, bulk and single-cell RNA sequencing of melanoma biopsies differentiated cells into two phenotypic states corresponding to two mutually exclusive gene expression signatures driven by the transcription factors MITF and AP-1/TEAD [18,163,164]. Further sequencing in additional melanoma cell lines and patient-derived xenograft (PDX) models unveiled a greater complexity beyond binary melanocytic–mesenchymal phenotypes [111,165,166,167]. Currently, at least seven in vivo melanoma phenotypes have been characterized by low to high MITF activity: undifferentiated (mesenchymal-like), neural crest stem cell-like (NCSC), interferon-active (IFN-active), starved melanoma cells (SMCs), intermediate, melanocytic, and hyper-differentiated. Fluctuating MITF activity during tumorigenesis and tumor progression confounds the ability to define its oncogenic role [164]. Additionally, MITF activity does not simply correlate with expression levels but also correlates with cell background, which dictates MITF protein stability [168], subcellular localization [169], protein interactors [170,171], microRNA regulation [172,173,174,175], DNA binding affinity [176], and DNA accessibility [64]. Furthermore, MITF activity depends on tumor microenvironment (TME) conditions, such as nutrient depletion [177,178], immune surveillance and inflammation [179,180,181], heterotypic interactions with normal cells [182], hypoxia [183,184], extracellular matrix (ECM) composition [185], and drug treatment [111,180,186,187]. Hence, an integrative adaptive response tuning MITF activity alters a cell’s probability of adopting a particular phenotypic state [167,188]. Genetic mutations drive tumorigenesis, and selected cancerous clones can overcome evolved host barriers against neoplastic growth. While genetic changes are irreversible, stochastic intra- and extra-cellular adaptive responses facilitate epigenetic plasticity, enabling cells to adopt reversible phenotypic states with temporospatial advantages (Figure 3).

Due to the expression of a mélange of melanocytic, mesenchymal, and neural-crest-like genes, starved and intermediate cells are considered poised state founders [111,167]. Starved cells present as metabolically hungry and arise following stressful conditions, such as hypoxia, nutrient depletion, and drug treatment. Undifferentiated, NCSC, and melanocytic states represent dormant cell reservoirs tolerant of cellular stress until the emergence of favorable microenvironmental conditions. Alternatively, cells can switch to highly proliferative intermediate or hyper-differentiated states which fuel tumor growth and replenish lesions with new clones. 

KDM5B, the histone demethylase which modulates melanoma response to immunotherapy via SETDB1 recruitment and ERV repression, has also been shown to regulate BRAF and MEK inhibitor resistance by controlling a shift between CD34^+^ and CD34^−^ subpopulations that vary widely in their treatment sensitivities [189]. BRAFi induces the enrichment of drug-resistant CD34^−^ melanoma cells, upregulation of KDM5B, and global reduction in H3K4me3. KDM5B’s enzymatic JmjC domain and enzyme-independent domains involved in chromatin binding both regulate the shift between drug-resistant and sensitive populations. Genetic and pharmacological inhibition of KDM5B diminishes the degree of phenotype switching from CD34^+^ to CD34^−^ during BRAFi treatment. KDM5B regulates transitions between melanoma-propagating cells with varying drug sensitivities that could potentially be exploited therapeutically. Additionally, the utility of CD34 expression as a prognostic biomarker for BRAFi sensitivity should be explored.

In addition to regulating immune checkpoint blockade resistance, SETDB1 may be a therapeutic target for patients resistant to combinatorial BRAF and MEK inhibition [190]. SETDB1 promotes H3K4 mono-methylation upstream of the thrombospondin 1 (THBS1) promoter, altering chromatin to an open and active state. THBS1 promotes melanoma aggressiveness and may facilitate therapeutic resistance through regulation of EMT phenotype switching [191,192,193]. Targeting of SETDB1 with a small-molecule inhibitor synergized with MAPK inhibition in sensitive cell lines and induced cell death in MAPKi-resistant melanoma cell lines.

**Table 2 cancers-14-05858-t002:** **Epigenetic regulators of targeted therapy**.

Gene	Function	Mechanism
SETDB1 [190]	Histone methyltransferase	SETDB1 inhibition is cytotoxic to BRAFi-resistant melanoma cells, and inhibition synergizes with BRAFi and MEKi.
SIRT6 [153]	Histone deacetylase	SIRT6 haploinsufficiency results in H3K56 acetylation at the IGFBP2 locus, activating PI3K/AKT/mTOR, and facilitating melanoma resistance to MAPKi. IGF-1R inhibition restores sensitivity to MAPKi. Complete SIRT6 loss activates DNA damage response due to global chromosomal instability and increases MAPKi sensitivity.
BMI1 [149]	Polycomb ring finger oncogene	BMI1 expression shifts melanoma to a metastatic state by inducing an invasive gene signature through WNT5A, ROR2, EGFR, and PDGFR, without a decrease in proliferation. WNT signaling maintains MITF expression, preventing proliferation defects typical of the invasive state, while concurrent EGFR and PDGFR upregulation maintains BRAFi resistance. BMI1 inhibition restored BRAFi sensitivity through WNT5A.
KDM5B [189]	Histone demethylase	KDM5B upregulation following BRAFi treatment decreases H3K4me3 levels and triggers the conversion of melanoma cells from drug-sensitive CD34^+^ to drug-resistant CD34^-^ cell states.

### 2.4. Non-Coding RNAs Involved in MAPKi Therapy Resistance

Non-coding RNAs (ncRNAs) are involved in several molecular pathways underpinning physiological and pathological conditions, including cancer [194]. ncRNAs can arise from active enhancers, promoters, intragenic and intergenic regions, or alternative splicing of canonical transcripts [195]. They are classified into two major groups according to their size and structure: (i) small non-coding RNAs, which are smaller than 200 bp and include microRNAs (miRNAs); and (ii) long non-coding RNAs (lncRNAs) which account for 15,000–90,000 annotated transcripts [196,197], are more than 200 nucleotides in length, and include the subclass of circular RNAs (circRNAs). The expression of non-coding RNAs (ncRNAs) is often altered in cancer by single-nucleotide polymorphisms (SNPs) and copy-number variations (CNVs). ncRNAs can interact with chromatin, proteins, and other RNAs involved in tumorigenesis [198]. 

Ninety percent of melanoma patients express lncRNA SAMMSON, which is undetectable in normal melanocytes [80]. SAMMSON induction is due to nearby MITF locus amplification and upregulation of the transcription factors SOX10 and SOX9, which directly bind the SAMMSON promoter. SAMMSON fosters melanoma proliferation and survival by bursting mitochondrial activity and biogenesis through the p32 pathway [81]. Similarly, melanoma cells can overexpress the lncRNA LENOX (LINC00518) following LENOX genomic amplification or increased activity of SOX10 and TFAP2A [82]. LENOX works in concert with SAMMSON to promote mitochondrial oxidative phosphorylation adaptation during tumor progression and upon MAPKi treatment and resistance. 

On the other hand, lncRNA TINCR is a negative translation regulator of the integrated stress response (ISR) transcription factor ATF4 through the direct binding of mRNAs driving the mesenchymal-like melanoma phenotype [199]. Even though TINCR regulation is not well-characterized, it is one of the few examples of ncRNAs indirectly regulating melanoma transcriptional states. In addition to lncRNAs, multiple miRNAs (e.g., miR-7 [200], miR-125a [103], miR-204-5p [201], and miR-211-5p [202]) have been found to be involved in resistance to MAPKi.

## 3. Concluding Remarks

Remarkable advances within the last decade have shattered the enduring untreatable paradigm of metastatic melanoma. Pharmacological inhibition of the MAPK pathway members BRAF and MEK or monoclonal antibodies against immune checkpoint markers CTLA-4 and PD-1 have provided metastatic melanoma patients with previously unattainable opportunities for meaningful and durable responses. Unfortunately, intrinsic or acquired resistance takes this benefit away for half of patients. An evolving understanding of cellular biology has driven scientific exploration beyond the limits of the genome and has reinforced the role of epigenetic mechanisms in regulating homeostatic and oncogenic cell states. Efforts to enhance patient responses have also shifted toward a deeper understanding of the epigenomic landscapes that facilitate treatment-sensitive cell states, the mechanisms to unlock them, and their downstream effectors.

We chose to focus a major portion of this review on the emerging role of viral mimicry in regulating melanoma response to immunotherapy, the chromatin writers and erasers co-opted by melanoma for its repression, and the interactions with the immune system elicited by its induction. The convergence of multiple mechanisms suggests the possibility that this phenomenon may be a major avenue of melanoma progression through immune evasion and, conversely, an exploitable susceptibility for treatment. However, our understanding of viral mimicry, which we define as an instance of beneficial cellular hypochondria, requires a deeper mechanistic dive into its components, such as dsRNA accumulation, interferon signaling, or the generation and presentation of neoantigens containing certain viral features. Most of the mechanisms enabling viral mimicry examined in this review focus on post-translational modifications of histone topography. Post-translational modifiers are well-described in the literature, but their nuclear localization has hindered their effective targeting. In addition, their ubiquitous expression in healthy and cancerous cells limits their usefulness as oncolytic targets. Since most of the chromatin regulators described here (LSD1, PRMT7, SETDB1, KDM5B, and ATF7IP) converge functionally, it will be crucial to identify additional cancer-specific factors the targeting of which may be less toxic. Proper identification of such targets mandates the analysis of clinical samples—a major limitation of the studies reviewed here, which focused on in vitro cell cultures or in vivo mouse models. The limited quantities and sizes of melanoma patient tissues available for experimentation demand collaboration amongst researchers to interrogate panels of markers, rather than assessing genes of interest on an individual basis.

The reviewed publications reiterate the notion that, without simultaneous inhibition of multiple pathways, overcoming targeted therapy resistance becomes a perpetual whack-a-mole game against emerging survival pathways. Additionally, these studies uphold that targeting the epigenome enables simultaneous activation or repression of multiple pathways. Of particular interest are the lessons learned from SIRT6, demonstrating that complete loss can be beneficial for therapy sensitization due to DNA damage responses following global chromosomal disarray, provided that targeting is sufficiently discriminatory between healthy and malignant cells to leave a therapeutic window. Additionally, BMI1 epigenetic regulation demonstrates that melanoma cell states are not as binary (invasive or proliferative) as typically described; paradoxical epigenomic states exist which enable MITF alterations without proliferation changes or shifts in therapeutic BRAFi sensitivity. Melanocytic, intermediate, mesenchymal-like, and mitotic states are detectable in drug-naïve tumors [203], while NCSC, SMC, pigmented, and IFN-active states arise upon drug treatment [111,167,187]. The emergence of an IFN-active state in PDX models elicits particular interest due to the absence of functional immune cells. Moreover, this state arises upon MAPKi treatment, reinforcing the molecular link between inflammation and therapy resistance. Accordingly, IFN-active melanoma cells overexpress the multi-drug resistance pump ABCG2 and interferon-related genes (e.g., PD-L1 and HLA-A, -B, and -C), irrespective of the presence of cytokines in vitro [187], conferring a potential cross-resistance to in vivo MAPKi and immunotherapy. Further exploration of a possible connection between epigenetic ERV de-repression and phenotype switching to the IFN-active state should be investigated. Finally, although mechanistically unclear, SETDB1 and KDM5B demonstrate pleiotropic effects in melanoma targeted therapy and immunotherapy which demand deeper mechanistic exploration.

## 4. Future Perspectives

Tumorigenesis and tumor progression rely in part on dysfunctional epigenetic regulation following genetic mechanisms, providing attractive therapeutic opportunities. Historically, natural compounds have been bountiful medicinal sources. As such, triptolide, a transcription factor II H inhibitor extracted from the *Tripteryugium wilfordii* plant [204], and lurbinectedin, an RNA Pol II degrader obtained from the sea squirt *Ecteinascidia turbinate* [205], are efficacious transcriptional inhibitors that impair cancer survival. Minnelide, a water-soluble triptolide prodrug, inhibits MYC and is undergoing Phase II clinical testing in advanced refractory adenosquamous pancreatic carcinoma [206]. However, these approaches frequently lack specificity and exert significant toxicity against healthy cells [207]. To enhance cancer specificity, de novo drug synthesis and medicinal chemistry have focused on disrupting cancer-specific genes or the unique protein–protein interfaces of epigenetic regulators. 

Extensive compound profiling combined with titration of molecular 3D structures using nuclear magnetic resonance (NMR) or cryogenic electron microscopy (cryo-EM) are guiding the development of successful synthetic drugs. PROteolysis TArgeting Chimeras (PROTACs) are bi-modular molecules containing a ubiquitin ligase E3 and a target-protein-specific domain for proteosome-dependent ubiquitination and degradation [208]. AU-15330 is the first effective PROTAC against BRG1, BRM, and PBRM1 of the SWI/SNF complex, which account for ~20% of mutations in human cancers, including cutaneous melanoma [209]. Due to cancer cells’ addiction to dysregulated SWI/SNF, AU-15330 has enhanced cytotoxicity against malignant cells. Additionally, direct inhibition of MYC, one of the most amplified and de-regulated oncogenic transcription factors, has been a great ambition and an even greater challenge [210]. MYCi361, a PROTAC that disrupts the MYC-MAX heterodimer and promotes MYC degradation, has proven to be synergistic with anti-PD1 immunotherapy in a prostate cancer mouse model, highlighting the exciting possibilities of PROTAC technology [211]. 

Alternatively, epigenetic regulators can be disrupted at the RNA level with antisense oligonucleotides (ASOs). DNA or RNA sequences of 15 to 25 base pairs, ASOs hybridize with complementary RNA to block activity or trigger RNAse-H-dependent degradation. Bearing locked nucleic acids (LNAs) at both ends, ASOs’ high stability facilitate their administration in the absence of nanoparticles or liposomal vehicles and enable targeting of coding and non-coding RNAs in any subcellular compartment. ASOs are being tested in clinical trials of orphan genetic diseases and have proven to be safe and sufficient to reduce the symptoms of Duchenne muscular dystrophy (DMD) and neuronal ceroid lipofuscinosis (CLN7) patients [212]. Systemic administration of ASOs results in liver and kidney accumulation, reducing uptake by other tissues [213,214,215], but is preventable by conjugation with other molecules or local injection to improve tissue-specific delivery [216,217]. ASOs are finding their way in cancer therapy, as witnessed by their rapid development and clinical testing over the last few years.

The targeting of epigenetic regulators has the potential to be used in a wide spectrum of cancers, irrespective of lineage identity and governing oncogenic networks. Alone or in combination with conventional treatments, PROTACs and ASOs may represent the future for personalized medicine in patients bearing oncogenic drivers traditionally perceived as undruggable.

## Figures and Tables

**Figure 1 cancers-14-05858-f001:**
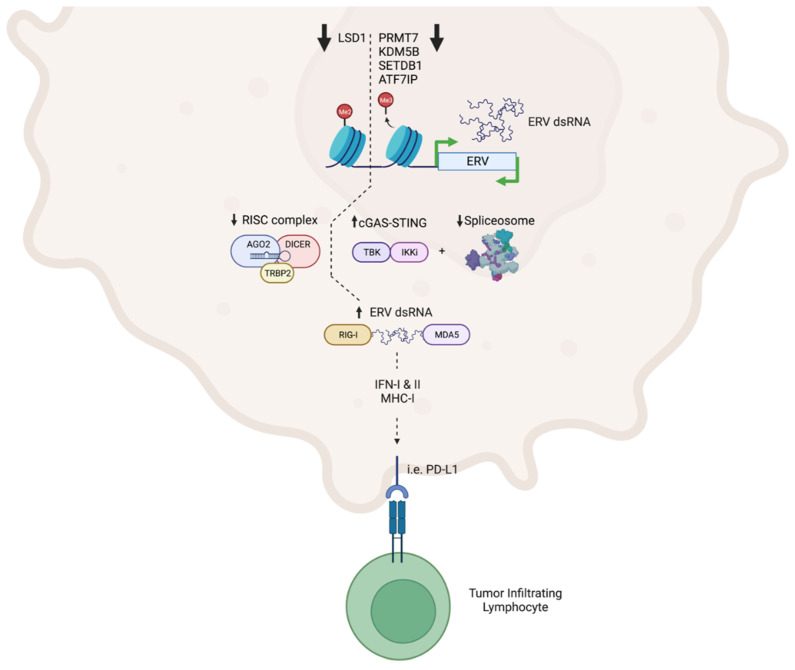
**Viral mimicry by epigenetic regulators: A beneficial instance of cellular hypochondria.** Epigenetic regulation of post-translational modifications by LSD1, PRMT7, KDM5B, ATF7IP, and SETDB1 modulates the expression of genomically integrated endogenous retroviral (ERV) elements and elicits interferon (IFN)-driven states that regulate anti-PD-1 and anti-CTLA-4 immunotherapy responses.

**Figure 2 cancers-14-05858-f002:**
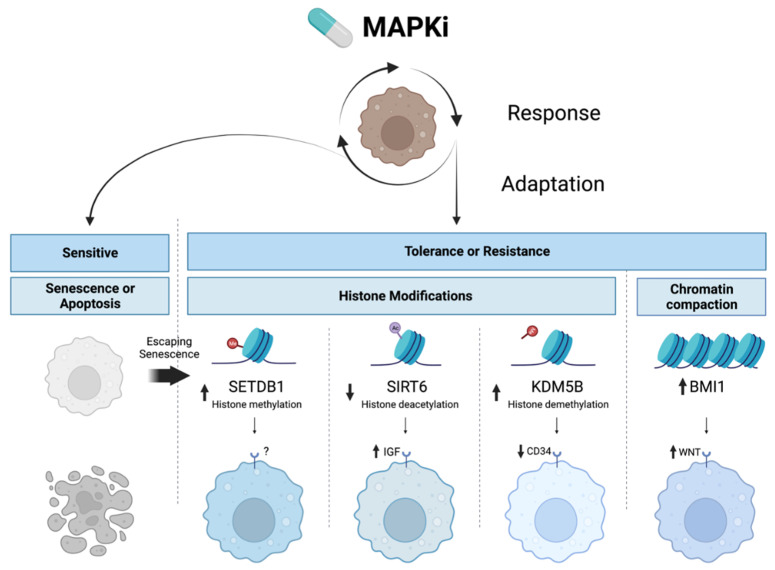
**Epigenetic mechanisms underlying MAPKi resistance in melanoma.** Inhibition of the MAPK pathway (MAPKi) results in the senescence or apoptosis of sensitive melanoma cell states. Meanwhile, escape from senescence and intrinsic and acquired resistance can be facilitated by histone modifications induced by SETDB1, SIRT6, and KMD5B or by chromatin compaction via BMI1, which activates cell survival pathways and changes melanoma cell states.

**Figure 3 cancers-14-05858-f003:**
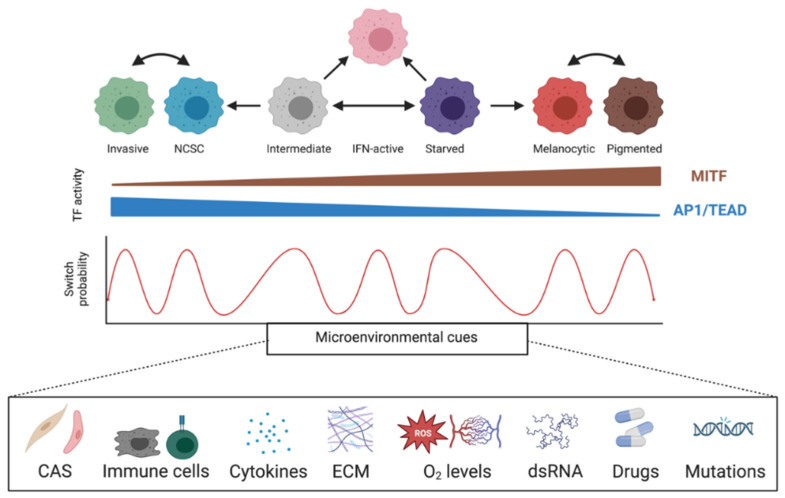
**Tumor microenvironment drives phenotypic diversity in melanoma.** The tumor microenvironment is characterized by the juxtaposition of heterotypic interactions between healthy tissue and melanoma cells, influencing two opposing MITF- or AP1/TEAD-dependent transcriptional programs that define distinct phenotypic states. Phenotype-switching represents a dynamic adaptation to microenvironmental variation comprising immune infiltration, inflammation, extracellular matrix (ECM), nutrient availability, oxygen levels, and melanoma-cell-autonomous mechanisms, such as de novo mutations and double-stranded RNA (dsRNA) regulation. CAS: cancer-associated stroma.

**Table 1 cancers-14-05858-t001:** **Epigenetic regulators of immune therapy through viral mimicry**.

Gene	Function	Mechanism
PRMT7 [136]	Protein arginine methyl transferase (PRMT)	PRMT7 inhibition represses DNMTs regulating ERV expression. ERVs cause dsRNA stress, detected by increased MDA5 and RIG-I due to H3K4me3 and H4R3me2 hypomethylation of their promoters. Shifting the cell into viral mimicry, the resultant IFN expression enhances antigen presentation and improves PD-1 & CTLA-4 checkpoint blockade via increased CD8^+^ T cell and reduced MDSC infiltration.
LSD1 [139]	Histone demethylase	LSD1 ablation increases ERV transcription and reduces core RISC complex proteins, generating cellular dsRNA stress detected by TLR3 and MDA5, inducing viral mimicry, activating IFN signaling, and increasing MHC-I and PD-L1 expression. Enhanced immunogenicity improves CD8^+^ T cell infiltration and boosts response to PD-1 checkpoint blockade.
SETDB1 [141]	Histone methyltransferase	SETDB1 loss de-represses TEs encoding viral antigens and immunostimulatory genes. Intact viral ORFs in induced TEs facilitate generation of MHC-I peptides and trigger CD8^+^ T cell responses. In B16F10-GVAX melanoma, SETDB1 knockout sensitized cells to PD-1 checkpoint blockade. Rare bi-directional TE transcription prevented the generation of dsRNA and interferon signaling and thwarted viral mimicry.
KDM5B and SETDB1 [143]	Histone demethylase and histone methyltransferase	KDM5B recruits and regulates SETDB1 in a proteosome-dependent manner. KDM5B loss increases bi-directional ERV transcription, accumulating dsRNA stress. MDA5 and cGAS depletion ablate IFN-I activation, MHC-I signaling, and induction of ISGs following KDM5B knockout. Increased CD8+ T cell activity following KDM5B knockout sensitized treatment-resistant YUMM1.7 melanoma cells to anti-PD-1 in vivo treatment.
ATF7IP and SETDB1 [144]	SETDB1 adaptor and histone methyltransferase	ATF7IP interacts with SETDB1 to repress ERV expression. ATF7IP deficiency reduces H3K9me3 deposition by SETDB1 and inhibits spliceosome activity, significantly increasing mRNA intron retention. Interferon signaling through IRF7 and IRF9 increase ERV antigen presentation, enhancing immunogenicity and facilitating clearance via elevated T cell infiltration.

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
