# Peer review of "Epigenetic Mechanisms Underlying Melanoma Resistance to Immune and Targeted Therapies"

_cancers, 2022, doi:10.3390/cancers14235858_

Round 1

Reviewer 1 Report (Previous Reviewer 2)

The authors have addressed most of the major concerns. 

Reviewer 2 Report (Previous Reviewer 3)

I would like to thank Author s for making appropriates changes in the manuscript. Congratulations.

This manuscript is a resubmission of an earlier submission. The following is a list of the peer review reports and author responses from that submission.

Round 1

Reviewer 1 Report

The review manuscript presented by Rubanov et al, entitled: “Epigenetic mechanisms underlying melanoma resistance to immune and targeted therapy” describes important mechanisms contributing to therapy resistance in melanoma.

The Authors extensively review epigenetic alterations and upregulation of survival pathways following targeted and immune-therapies against this aggressive disease. The review effectively points out the progress and challenges regarding the treatment of melanoma, and propose the targeting of epigenetic regulators as the future of personalized medicine against cancer.

The review is comprehensive, well written and a great summery of the present knowledge in the field of melanoma research.

Reviewer 2 Report

This manuscript is well written and best introduces the knowledge of epigenetics in Melanoma. However, several relevant features must be added and edited before it is ready for submission:

The manuscript lacks important epigenetic mechanisms. The contribution of DNA methylation (DNMTs and TETs), miRNA (study by Shani Dror et al. and Manuela Ferracin, should be added), non-coding RNAs, lncRNAs (seminal studies from Jean-Christophe Marine’s group and others), and RNA methylation in immunologic characteristics and antitumor properties of melanoma cells is completely lacking.

The authors mention that – “We have chosen to focus a major portion of this review on the emerging role of viral mimicry in regulating melanoma response to immunotherapy”, yet they have hardly described the types/families of ERVs (eg. Studies from Bunker CB and Zsuzsanna Izsvák’s group).

The section on interferon-mediated viral mimicry facilitated by ERV de-repression should include seminal studies indicating the role of DNMT (Chiappinelli et al., 2015) and CDK9 (Pandey et al, Hanghang et al, ) in regulating ERV expression and how they could be used to sensitize tumor models to immune checkpoint therapy. Toyocamycin, a specific CDK9 inhibitor has been shown to induce ERV expression and reduce the growth of resistant melanoma cells.

Figure 1 should be expanded to include above mentioned epigenetic regulators relevant to melanoma resistance and how each mechanism for example- DNMT, TETs, KDM, KMT, non-coding RNAs interact with one another.

Brief information is mentioned about the chromatin remodelers (Patrick Laurette, et al. – BRG1/BPTF), HATs (studies involving the role of p300 and melanoma growth, etc. should be included), HDACs (SIRT6, HDAC6, etc.).

The melanoma tumor microenvironment has not been described well. Studies from Hector Peinado, Raza Zaidi, and Glenn Merlino, should be added.

The authors have mentioned Triptolide and its efficacy. The fact that Minnelide (derivative of Triptolide) is currently in Phase I and Phase II clinical trials. This information should be added.

Minor comments:

Correct the following sentence – “Pharmacological inhibition or CRISPR/Cas9-mediated knockout of PRMT7 hypomethylates ERVs via DNMTs, accumulating dsRNA, a common characteristic of viral infections.” PRMT7 inhibition causes hypomethylation due to reduction of DNMT which then results in dsRNA….

Reviewer 3 Report

The review is focused more on epigenetic mechanism in melanoma particular to immune therapies. Few comments if author can add up to the review.

- Can author add more clear figure1?

- Plenotype switching is one the mechanism responsible for the drug resistance and immune escape. can author add bit more details available mechanism?

Rest sonds good. Thank you for sending me the manuscript.
